# Untrained perceptual loss for image denoising of line-like structures in MR images

**Elisabeth Pfaehler**[1]*, **Daniel Pflugfelder**[2], **Hanno Scharr**[1]

**1** Institute for Advanced Simulation: Data Analytics and Machine Learning (IAS-8), Forschungszentrum Jülich GmbH, Germany, Jülich, North-Rhine-Westfalia, Germany, **2** Institute of Bio- and Geosciences: Plant Sciences (IBG-2), Forschungszentrum Jülich GmbH, Germany, Jülich, North-Rhine-Westfalia, Germany

* e.pfaehler@fz-juelich.de

**Data availability statement:** Not all roots data can be shared publicly because some of the images were acquired within industrial projects that do not allow for free data sharing. However,

## Abstract

In the acquisition of Magnetic Resonance (MR) images shorter scan times lead to higher image noise. Therefore, automatic image denoising using deep learning methods is of high interest. In this work, we concentrate on image denoising of MR images containing line-like structures such as roots or vessels. In particular, we investigate if the special characteristics of these datasets (connectivity, sparsity) benefit from the use of special loss functions for network training. We hereby translate the Perceptual Loss to 3D data by comparing feature maps of untrained networks in the loss function. We tested the performance of untrained Perceptual Loss (uPL) on 3D image denoising of MR images displaying brain vessels (MR angiograms - MRA) and images of plant roots in soil. In this study, 536 MR images of plant roots in soil and 450 MRA images are included. The plant root dataset is split to 380, 80, and 76 images for training, validation, and testing. The MRA dataset is split to 300, 50, and 100 images for training, validation, and testing. We investigate the impact of various uPL characteristics such as weight initialization, network depth, kernel size, and pooling operations on the results. We tested the performance of the uPL loss on four Rician noise levels (1%, 5%, 10%, and 20%) using evaluation metrics such as the Structural Similarity Index Metric (SSIM). Our results are compared with the frequently used L1 loss for different network architectures. We observe, that our uPL outperforms conventional loss functions such as the L1 loss or a loss based on the Structural Similarity Index Metric (SSIM). For MRA images the uPL leads to SSIM values of 0.93 while L1 and SSIM loss led to SSIM values of 0.81 and 0.88, respectively. The uPL network's initialization is not important (e.g. for MR root images SSIM differences of 0.01 occur across initializations, while network depth and pooling operations impact denoising performance slightly more (SSIM of 0.83 for 5 convolutional layers and kernel size 3 vs. 0.86 for 5 convolutional layers and kernel size 5 for the root dataset). We also find that small uPL networks led to better or comparable results than using large networks such as VGG (e.g. SSIM values of 0.93 and 0.90 for a small and a VGG19 uPL network in the MRA dataset). In summary, we demonstrate superior performance of our loss for both datasets, all noise levels, and three network architectures. In conclusion, for images

around 60% of the roots images will be shared what will be enough to reproduce the study findings. We will share this data in the following weeks. We will start now with preparing the images and select the ones that can be shared. We will upload the data on Jülich data: https://data.fz-juelich.de/. All implemented python code is now being prepared and will be made publicly in the next two/three weeks on github.

**Funding:** This work was supported by the Networking Funds of the Helmholtz Association of German Research Centres 276 (HighLine ZT-I-PF-4-042).

**Competing interests:** The authors have declared that no competing interests exist.

containing line-like structures, uPL is an alternative to other loss functions for 3D image denoising. We observe that small uPL networks have better or equal performance than very large network architectures while requiring lower computational costs and should therefore be preferred.

## Introduction

Enhancing image quality of 3-dimensional (3D) images by suppressing image noise is important in bio-medical applications. Conventional methods such as spatial filtering or other, edge-preserving techniques (e.g. bilateral filtering [1,2]) are often used for image denoising [3,4]. However, these conventional filtering methods tend to lead to blurry results. In recent years, convolutional neural networks (CNNs) overcame the limitations of conventional denoising methods and showed strongly improved results for both 2D and 3D images [5,6]. For image denoising using deep learning, several network architectures have been proposed [5,7,8]. The majority of works reporting on image denoising concentrate on the optimal network architecture. However, not only network structure but also selecting an appropriate loss function is important for network performance. [9,10] As the loss function is a crucial part of neural network training, the focus of this work lies in finding the optimal loss function for 3D image denoising of images containing line-like structures.

For Magnetic Resonance (MR) images, image quality is related to scan time, i.e. shorter scan times result in images with higher image noise [11]. While causing higher noise levels, shorter scan times lead to better patient comfort and higher daily patient throughput. Image denoising can transform these shorter scans to images with similar image quality as scans acquired during a longer time. Please note that in contrast to other works, this paper focus on image-image denoising and is not considering image reconstruction [12,13].

For 3D MR image enhancement, two approaches are frequently used: (1) In the majority of the works, the images are denoised slice by slice, i.e. using a 2D network [14–17]. (2) Other works denoise the whole 3D volume (3D networks) [18–20]. In [21,22] the authors compare the 2D slice-by-slice with the pure 3D approach using various denoising networks. They demonstrate that using the 3D volume and 3D networks leads to consistently better results than the 2D approach as 2D denoising methods ignore relevant 3D information. Therefore, in this work, we concentrate on 3D networks.

For 2D images, the 'Perceptual Loss' [23–29] (also called 'content representation' [30]) is frequently used as loss function for image enhancement and outperformed traditional pixel-by-pixel losses. The Perceptual Loss is also often used in combination with L1- or L2-loss where it also leads to performance improvements [31]. The standard Perceptual Loss (sPL) minimizes differences between features extracted from reconstructed and ground truth image using a pre-trained neural network. The success of the sPL has been thought to lie in the fact that the loss network extracts meaningful features, as it has been trained on a large variety of images [24,28]. Such pretrained networks are unavailable in 3D hampering the application of sPL for 3D MRI enhancement. However, in recent works an untrained Perceptual Loss (uPL) was used in 2D image super-resolution and compared features from untrained, randomly initialized networks [32,33]. The randomly initialized loss networks were exclusively used to extract feature maps from high-quality images and remain constant during the super-resolution training process and performed similarly to sPL [33]. This opens a door towards using Perceptual Losses also in 3D.

For 3D image enhancement, the most frequently used loss functions are L1 or L2 loss which compare generated and ground truth images voxel-by-voxel and thereby ignoring voxel neighborhoods [21,34]. As displayed in Table 1, most recent studies adressing image denoising of MR brain or MRA images focus on the most adequate network architecture while using conventional loss functions such as L1 and/or SSIM-loss. Therefore, the aim of this work is to investigate if the uPL can be used beneficially for 3D images containing line-like structures. The research objective is to assess the impact of uPL network parameters on the results. We are especially interested if also small, simple networks can be used in the uPL.

**Our contribution:** In summary, our contributions are as follows:

- We propose 3D uPL with small 3D convolutional networks in the loss function for 3D denoising - a straightforward translation from the 2D case.
- We demonstrate the suitability of uPL for 3D denoising of MR images containing line-like structures outperforming competing losses by a large margin.
- We investigate and discuss the influence of network hyper-parameters like network depth, layer thickness, kernel size, and pooling operations.
- We investigate the suitability of the uPL for three denoising network architectures and demonstrate its superiority for all the three architectures, where we also provide a novel 3D implementation of the Restormer [5] network.

## Materials and methods

### Datasets

We included two datasets in this study. The first dataset consists of 536 MR images of plant roots in soil with voxel size 0.5 mm × 0.5 mm × 0.99 mm and a typical image size of 192 × 192 × 1000. For the root dataset, 380 images were used for training, 80 images for validation, and 76 for testing.

**Table 1. Recent networks used for MR image denoising and MRA data. As displayed, all listed works use either conventional loss-functions such as L1 and/or SSIM loss or a 2D version of the Perceptual Loss.**

| Authors | Dataset type | Purpose | Network architecture | Loss function |
|---|---|---|---|---|
| Tian et al. | MR Brain (T1) | Denoising | Modified U-Net | L1 |
| Haetsam et al. | MR Brain (T1, T2) | Denoising | Modified U-Net | L1 |
| Ran et al. | MR Brain | Denoising | 2D Generative Adversarial Network | 2D Perc. Loss |
| Wu et al. | MR Brain | Denoising | Deep Network (no downsampling) | L2 |
| Li [35] et al. | MR Brain | Denoising | ParallelNet (no downsampling) | L1 |
| Gurrola et al. [36] | MR Brain | Denoising | Modified Autoencoder | L1 |
| Lee et al. [37] | MR | Denoising | Diffusion Probabilistic | L1 |
| [37] | Animal images | | Model | |
| Fiasam et al. [38] | MR Brain | Denoising | Encoder/Decoder with attention modules | L1 |
| You et al. [39] | MR Brain | Denoising | CNN | L1 |
| Kuestner et al. [40] | MRA | Super-Resolution | GAN | L1+SSIM |
| Wicaksono et al. [41] | MRA | Super-resolution | GAN | L1+SSIM |

The MRA data consists of MR images of brain vessels. This dataset is part of the publicly available IXI datasets [1]. We had no access to any information that could identify the scanned individuals. The typical MRA image size of this dataset is $512 \times 512 \times z_{dim}$. The $z_{dim}$ varies from patient to patient. The training set was randomly split into 300 training, 50 validation, and 100 test images.

An example image for each dataset is displayed in Fig 1. For both datasets, four levels of Rician noise were artificially added to assess the impact of the loss functions for different signal-to-noise ratios (1%, 5%, 10%, and 20% noise added). An example root and MRA image

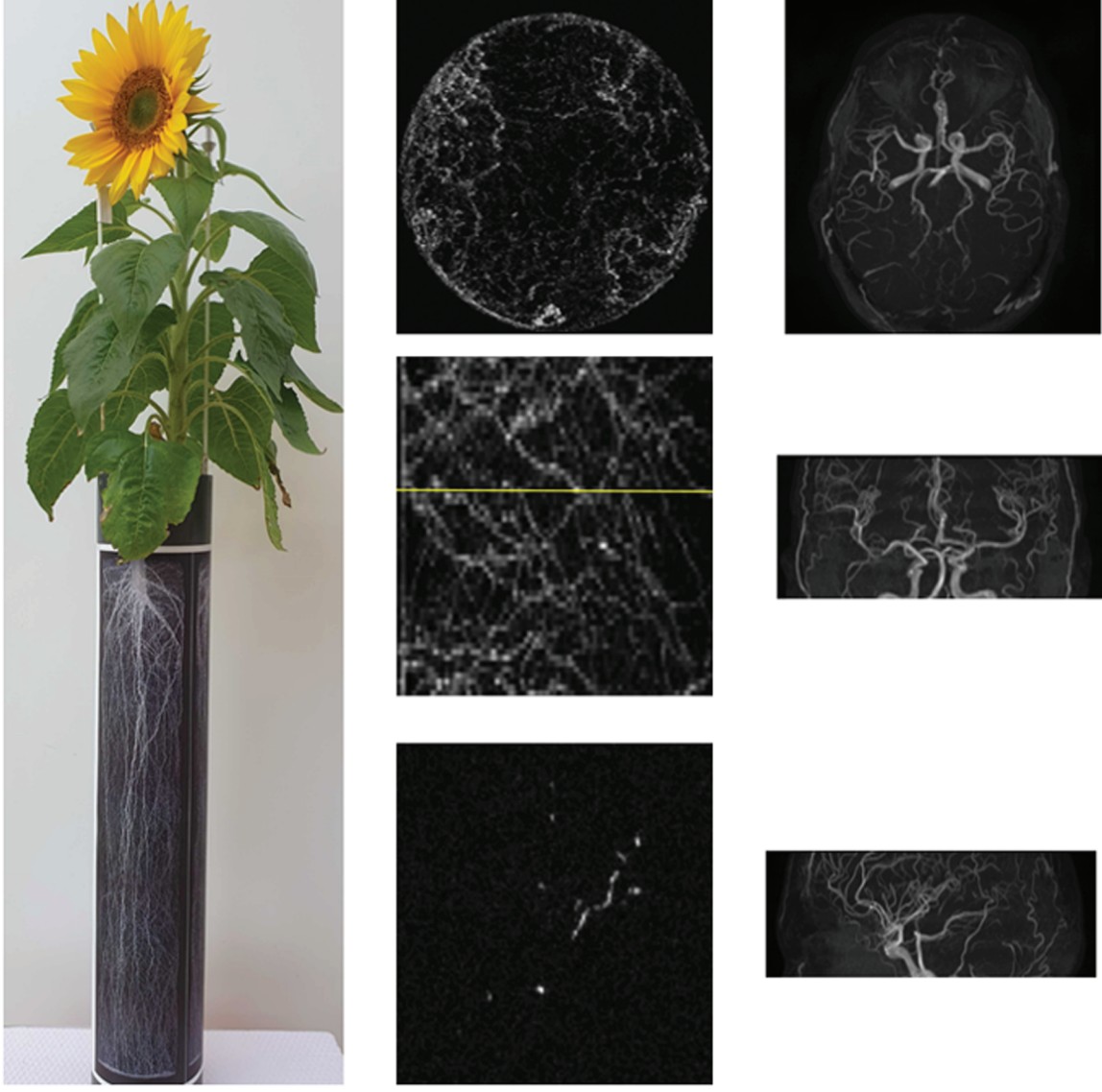

**Fig 1. Example 3D MR images of plant roots and brain vessels.** First column: Plant scanned in the MR system, second column: maximum intensity projections (MIP) of MR root image (top: axial plane, mid.: sagittal plane) and a single slice of the 3D image marked yellow in the sagittal plane (bot.). Next columns, top to bot.: Axial, coronal, and sagittal MIP of an MRA image.

[1]   https://brain-development.org/ixi-dataset/, assessed on 09/10/2022.

with different noise levels is displayed in Fig 2. Experiments regarding loss network structure were performed for noise level 3, i.e. 10%.

## Denoising networks

Three network architectures are compared in this study: a DnCNN [42], a ResNet architecure [28] and a Transformer network as proposed for 2D by Zamir et al. [5] translated to 3D. Please note that we also tested U-Net like architectures. However, especially for our very fine root data, these architectures generally led to much worse results than the shown methods, for pixel-wise losses as well as the better but still bad performing uPL. Therefore, we concentrate in this work on the three mentioned architectures.

The DnCNN consists of one convolutional layer with 64 feature maps and kernel size 3 followed by a ReLU activation function. Next, are three blocks consisting of a convolutional layer with 64 feature maps and kernel size 3, a ReLU activation, and a batch normalization layer. The last layer is a convolutional layer with kernel size 3 and 1 output channel.

The ResNet consists of one convolutional layer followed by a PreLU activation function. The first convolutional block is followed by five residual blocks, where each residual block contains one convolutional layer, a batch normalization layer, and a PreLU activation function. The residual blocks are connected via a skip connection. The residual blocks are followed by a convolutional layer, a batch normalization, and another PreLu activation.

The Transformer network [5] is specially designed such that it can be used for large images while modeling global connectivity. In this network, multi-head 'transposed' attention (MDTA) blocks are introduced applying attention across feature dimensions rather than

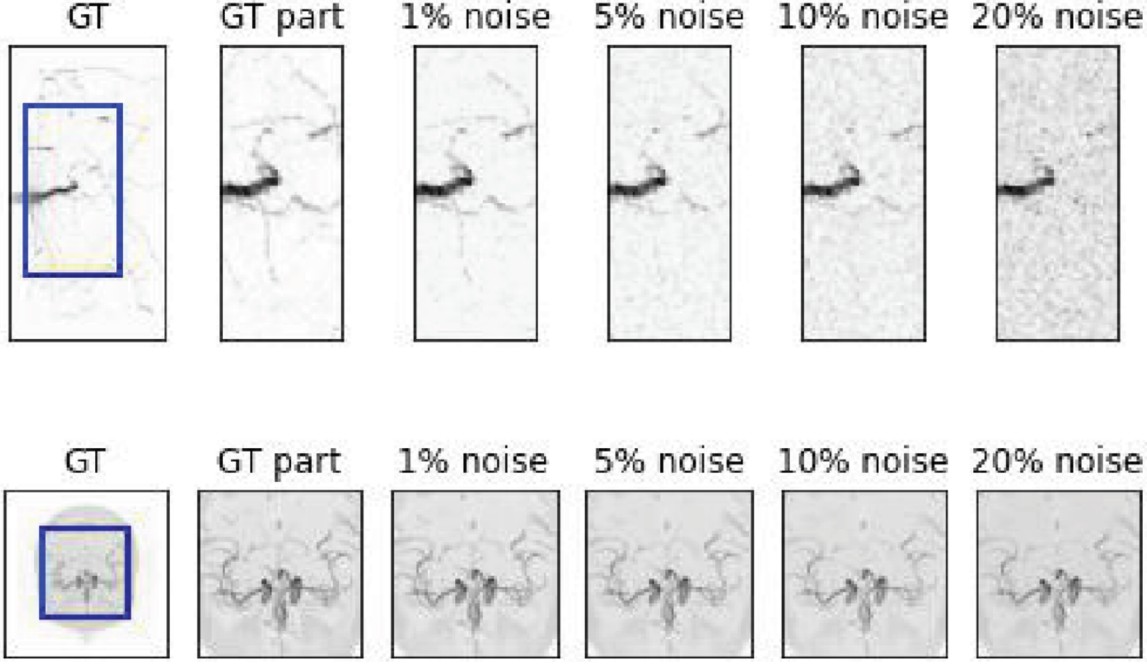

**Fig 2. Maximum Intensity Projections of original and noisy images.** From left to right: Original image, cropped part of original image, cropped part of image with 1%, 5%, 10%, and 20% added noise.

across spatial dimensions. The used Transformer networks starts with a convolutional layer with kernel size 3 and 8 output feature maps. The convolutional layer is followed by a MDTA block. Before feeding the data to the MDTA blocks, the images are resized to the size (image height · image width · image depth) × number of channels. After the MDTA blocks follow feed-forward blocks consisting of two convolutional layers (with kernel size 3 and 16, and 32 output feature maps) and a gating layer. As for our fine structures, UNet-like architectures lead to performance drops, the Transformer blocks were in our study combined sequentially, in contrast to the original 2D Restormer [5]. A graphical overview of all network structures is displayed in the (S1, S2, and S3 Figs).

The performance of the proposed uPL is compared with the L1 loss across the mentioned network architectures and different noise levels. For the evaluation of characteristics of the uPL network, the DnCNN is used as example network.

## Loss functions

In the next paragraphs, we first define the uPL. We then compare the denoising results using an uPL with a small loss network with conventional loss functions used in the literature. This experiment serves as a simple demonstration of the suitability of the uPL for image denoising tasks. In the next section, we then investigate the impact of different characteristics of the uPL network on the denoising results. All experiments were performed with five different random seeds.

**Untrained Perceptual Loss.** Perceptual Loss (PL) minimizes the differences between feature maps of neural networks of generated and ground truth images. PL is defined in 1

$$PL = \sum_j \frac{1}{C_j H_j W_j D_j} ||\phi_j(\hat{y}) - \phi_j(y)||_2^2 \tag{1}$$

where $C_j, H_j, W_j, D_j$ refer to the number of channels, the image height, the image width, and the image depth in layer $j$, respectively. $\hat{y}$ refers to the generated image and $y$ to the corresponding ground truth image, and $\phi_j$ outputs the activation of the $j^{\text{th}}$ layer in the loss network.

**Proof of concept - Comparison with conventional loss functions.** As proof of concept, we compared the results when using a small uPL network in the loss function with the most frequently used loss functions for image enhancement. We compare (1) the L1 loss [21,34,43], (2) a loss maximizing SSIM [44,45], (3) uPL with the untrained 3D version of VGG19, (4) uPL with the untrained 3D AlexNet, and (5) uPL with a simple loss network with three convolutional layers (each with 32 features and kernel size 3, ReLU activation).

We use this proof of concept study and the small network as a starting point to investigate which impact different characteristics of an uPL network have on the results. In particular, we concentrate on the following questions: *Can small and simple networks be successfully used in uPL? What are the important architectural parameters for successful use in uPL? What impact has the weight initialization? Is there a network that works best for all datasets or does the best network structure depend on image content? Is the impact of the uPL dependent on the denoising architecture?* As there are more hyper-parameters than we can reasonably consider in the uPL, we concentrate on a few prominent ones. Our goal is to find an easy working solution rather than infeasibly covering the entire parameter space. We tested the performance of different uPL network characteristics for 10% added noise and the DnCNN architecture. If a network characteristic leads to a clear improvement, this network characteristic was used as the default characteristic. If performance across similar characteristics was similar, we used the simplest solution (i.e. the smallest network) in the consecutive tests.

### Characteristics of networks used in uPL.

**Initialization of network used in uPL.** As the networks in the uPL are not trained, different initializations may have considerable impact on the performance. We investigate the impact of weight initialization in the uPL network by analyzing the results using five different random seeds for weight initialization in PyTorch. Weights were drawn from a uniform distribution $\mathcal{U}(-\sqrt{k}, \sqrt{k})$ where $k = (C_{in} * \prod_i K_i)^{-1}$ and $K_i$ is the size of the $i^{th}$ kernel. The loss network was a simple loss network with three convolutional layers, kernel size 3, and 32 feature maps.

Regarding the initialization method, five initialization methods were tested: (1) Kaiming uniform, (2) Kaiming normal, (3) Xavier uniform, (4) Xavier normal, and (5) the default uniform initialization of PyTorch (i.e. weights are drawn from a uniform distribution) [46,47].

**Depth of network used in uPL.** Previous works demonstrated that for 2D images deeper loss architectures resulted in better performance for image segmentation and super-resolution tasks [28,32]. We investigate if this holds also for 3D data, by varying uPL network depth. We tested networks with depths 3, 5, 7, 9, and 13 convolutional layers connected with a ReLU activation function. Additionally, we tested kernel sizes of 3, 5, 7, and 9. As default, each convolutional layer created 32 feature maps.

**Impact of pooling operations of network used in uPL.** Down-sampling a signal creates aliasing and potential loss of information when fine details or other high spatial frequency signals are present. This is why down-sampling or pooling is recommended to be applied after sufficient smoothing only (see e.g. [48]). As noise-initialized kernels typically have poor smoothing behavior the use of pooling layers in the uPL might lead to a loss in information and therefore to a performance drop. To assess the impact of pooling layers, we use an uPL network with five convolutional layers and added max-pooling operations. We were interested if the number of pooling operations impacts the results. The number of pooling operations varied from 1 to 3. We started with only one pooling operation after the first convolutional layer. Next, we added a second pooling layer after the second convolutional layer and a third pooling operation after the third convolutional layer.

### Implementation and training details

All experiments were implemented in Pytorch 1.10.0 and training was performed using PyTorch lightning. Networks were trained for 30.000 iterations with batch size 16, Adam optimizer [49] and learning rate 0.001. The training parameters were chosen as they lead to the overall best performance in the validation sets. For this purpose, networks were trained for 10.000, 15.000, 20.000, ... 50.000 iterations, batch size was set to 4, 8, 16, 24, and 32, and the learning rate was set to 0.001,0.002, ... 0.005. All possible combinations of these parameters were investigated and the parameters leading to the best performance in the above defined validation datasets were selected. For training, images were cropped randomly to a size of $96 \times 96 \times 96$. All codes will be available online after paper publication.

### Evaluation metrics

For evaluation, we calculate the structural similarity index (SSIM), peak signal-to-noise ratio (PSNR), and mean squared error (MSE) between generated and ground truth images. Mean and standard deviation (std) values over the datasets and random seeds are reported if not stated differently. As both datasets contain fine structures, the evaluation metrics were additionally calculated on the parts of the image containing important information. A cube of size $52 \times 52 \times 52$ was cropped from the root data. As the roots grow from top to bottom and

because the plant is always placed in the center of the scanner, the image was cropped such that the upper, middle image part was always present in the evaluation. I.e. the image was cropped from z-slices 0 - 52, and x-, and y-slices from 70-132. A cube of size $68 \times 68 \times 68$ is cropped from the middle of the MRA images as these images contain important information mainly in the image center i.e. the center of the image was determined and a cube with the mentioned size was cropped such that the center of the cube aligned with the center of the image. An example illustration is displayed in S4 Fig.

## Results

### Proof of concept - Comparison with conventional loss functions

In this experiment, the performance of different loss functions frequently used for image denoising are compared with an uPL containing a small network (three convolutional layers, kernel size 3). In this experiment, we observe consistent results for both datasets see Table 2. For both datasets, uPL using a small network leads to the best evaluation metrics compared with the other loss functions included in this study. For example, uPL results in an SSIM of $0.86 \pm 0.01$ and a PSNR of $38.1 \pm 0.4$ for the roots dataset. For comparison, L1 (SSIM) loss yields a mean SSIM of $0.79 \pm 0.02$ ($0.63 \pm 0.03$) and a PSNR of $37.4 \pm 0.5$ ($31.7 \pm 0.8$). For the MRA dataset improvements are similar. uPL yields an SSIM of $0.9 \pm 0.01$ while the L1 loss yields an SSIM of $0.79 \pm 0.02$ and SSIM as loss function leads to an SSIM of $0.86 \pm 0.04$.

The uPL containing a small network also outperforms the straight-forward 3D extension of uPL using AlexNet or VGG. For the MRA data, the performance drops to an SSIM of $0.86 \pm 0.02$ (PSNR $30.9 \pm 0.3$). The same holds for the root images, where the drop in performance is high. In contrast, using VGG19 in the loss leads to a slight performance drop when compared with the small network for both datasets. However, differences are not pronounced and almost not visible when comparing images by eye.

Fig 3 shows denoising results of one MR root and one MRA image for 10% noise added reconstructed with the DnCNN. As displayed, using L1 loss or SSIM loss leads to strong suppression of fine roots and to incorrect reconstruction of finer vessels. Also uPL using AlexNet fails to reconstruct finer roots and vessels accurately. The DnCNN trained with the smaller network (3 convolutional layers and kernel size 3), the VGG19 network creates root and MRA images where also fine details are correctly reconstructed.

### Impact of network initialization

To demonstrate that the superior performance of our small loss network is not due to chance, we investigated the performance for five different random seeds in the loss network.

**Table 2. Mean and standard deviation values for the five random seeds used for network training for MRA images and MR root images for the loss functions included in this study. Best results are underlined. Mean and standard deviation values for the evaluation metrics calculated exclusively on the center image parts as well as std values across test sets are given in the Supporting information S2 and S3 Tables.**

| | MRA | | | MR root | | |
|---|---|---|---|---|---|---|
| Loss | SSIM | PSNR | MSE | SSIM | PSNR | MSE (roots) |
| L1 | $0.79 \pm 0.02$ | $31.4 \pm 0.41$ | $0.0047 \pm 0.004$ | $0.79 \pm 0.02$ | $34.4 \pm 0.5$ | $0.038 \pm 0.006$ |
| SSIM loss | $0.86 \pm 0.04$ | $35.6 \pm 1.1$ | $0.0051 \pm 0.011$ | $0.63 \pm 0.03$ | $31.7 \pm 0.8$ | $0.058 \pm 0.008$ |
| VGG19 | $0.87 \pm 0.01$ | $40.2 \pm 0.3$ | $0.0075 \pm 0.0008$ | $0.84 \pm 0.01$ | $37.8 \pm 0.3$ | $0.031 \pm 0.007$ |
| AlexNet | $0.86 \pm 0.02$ | $30.9 \pm 0.3$ | $0.0043 \pm 0.001$ | $0.75 \pm 0.01$ | $28.4 \pm 0.4$ | $0.043 \pm 0.006$ |
| Our SimpleNet | $0.90 \pm 0.01$ | $41.3 \pm 0.4$ | $0.0043 \pm 0.0004$ | $0.86 \pm 0.01$ | $38.3 \pm 0.1$ | $0.031 \pm 0.005$ |

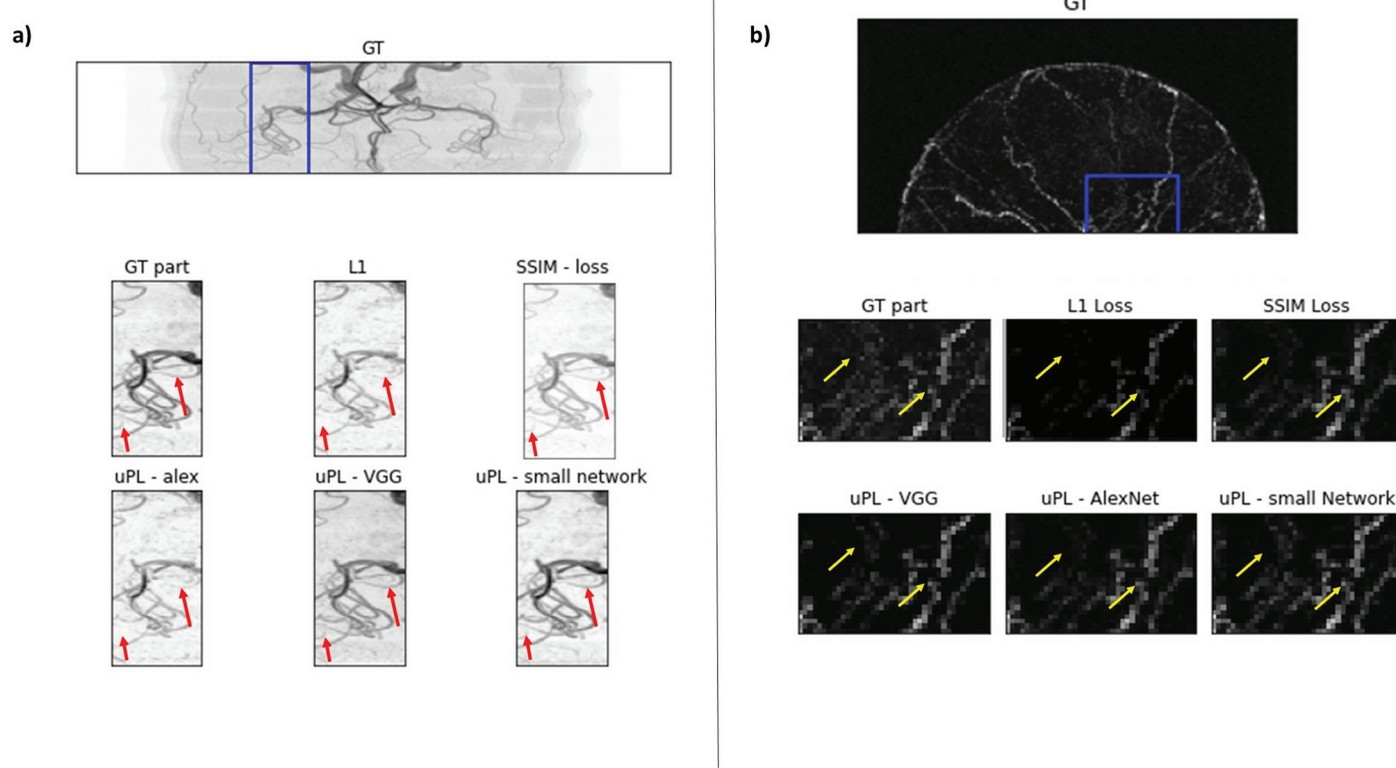

**Fig 3. MIP of example 3D MR images of plant roots and brain vessels: Ground truth image and image parts after denoising the image that was disturbed with 10% noise.** Upper row: MIP of whole image. In the second and third row, the marked region is zoomed in. Zoomed part reconstructed with different loss functions. Differences are clearly visible and remarkable differences are marked with an arrow.

Additionally, the impact of network initialization on performance metrics is investigated. For both datasets, the differences across weight initialization seeds are small (see Supporting information Table S1). E.g. for the MRA dataset different seeds resulted in PSNR between 41.2 and 41.4 and SSIM between 0.90 and 0.91. For the root dataset, different seeds resulted in SSIM values between 0.85 and 0.86.

All but the uniform initialization method lead to comparable results across datasets (see Table 3). For the MRA dataset, uniform initialization lead to SSIM performance drop from e.g. 0.92 for Xavier normal to 0.86 to uniform initialization. For the root dataset, no difference

**Table 3. Mean and std evaluation metrics for MRA and MR root images for different initializations of the untrained loss network.** As shown, different initializations have a minor impact on the results.

| uPL Loss initialization method | MRA | | | MR root | | |
|---|---|---|---|---|---|---|
| | SSIM | PSNR | MSE | SSIM | PSNR | MSE (roots) |
| Kaiming uniform | 0.91 ± 0.01 | 40.6 ± 0.3 | 4.3e-3 ± 2e-4 | 0.86 ± 0.01 | 38.8 ± 0.2 | 0.034 ± 0.001 |
| Kaiming normal | 0.91 ± 0.01 | 40.5 ± 0.2 | 4.2e-3 ± 1e-4 | 0.85 ± 0.02 | 38.8 ± 0.3 | 0.034 ± 0.002 |
| Xavier uniform | 0.92 ± 0.01 | 40.2 ± 0.2 | 4.0e-3 ± 2e-4 | 0.85 ± 0.01 | 38.8 ± 0.2 | 0.032 ± 0.001 |
| Xavier normal | 0.92 ± 0.00 | 40.4 ± 0.3 | 4.3e-3 ± 4e-4 | 0.86 ± 0.01 | 38.9 ± 0.2 | 0.041 ± 0.001 |
| Default uniform | 0.86 ± 0.01 | 36.7 ± 0.3 | 4.9e-3± 5e-4 | 0.85 ± 0.01 | 38.8 ± 0.3 | 0.05 ± 0.002 |

## Number of convolutional layers vs. kernel size

Results in Table 4 show that the number of convolutional layers and kernel sizes had a minor impact on the results.

For the MRA images, seven convolutional layers with kernel size 3, five convolutional layers with kernel size 3 or 5 lead with an SSIM of 0.93 to the best results. Similarly, for the root dataset, five convolutional layers with kernel size 5 led with an SSIM of 0.86 to the best results, followed by three convolutional layers with kernel size 3. Taking the variance over the five random seeds into account, the difference across numbers of layers and kernel size is small. The kernel size has little impact on the results for both dataset. A smaller network depth is beneficial.

## Impact of pooling layers

As pooling operations reduce image details, we investigate if the use of pooling operations might have an impact on the results. For the root images, pooling operations hurt performance slightly (e.g. SSIM without pooling 0.87, with one max pooling operation 0.85). For the MRA dataset, max pooling operations increase evaluation metrics for 1 and 2 pooling operations (SSIM 0.96 and 0.94). However, evaluation metrics dropped slightly for 3 pooling operations (SSIM 0.91). This impact is illustrated in Fig 4.

## Denoising network architecture vs. loss function for different noise levels

Lastly, we investigated the impact of the denoising network architecture across noise levels. Hereby, we were interested in the benefits of uPL across network architectures. We investigated the performance for the previously described DnCNN, ResNet, and Transformer architectures across noise levels. We used a network with three convolutional layers and kernel size 3 in the uPL.

Table 5 lists the SSIM for both datasets and all noise levels. Example results are displayed in 5 The improvement when using uPL depends on the network architecture and the noise level. Occasionally, uPL performed equal or slightly worse than L1 loss. For the Transformer network, the uPL showed its benefit, especially for higher noise levels (SSIM of $0.78 \pm 0.02$

**Table 4. Mean and std SSIM values for different kernel sizes and network depth for MRA (above) and for the MR root dataset (below). All other evaluation metrics can be found in the Supporting information S4 and S5 Tables. As displayed, differences across network sizes and kernel sizes are small.**

| Metric | Number of conv. layers | | | |
|---|---|---|---|---|
| Kernel size | 3 conv | 5 conv | 9 conv | 13 conv |
| | SSIM - MRA dataset | | | |
| 3 | $0.90 \pm 0.01$ | $0.93 \pm 0.01$ | $0.90 \pm 0.02$ | $0.90 \pm 0.01$ |
| 5 | $0.92 \pm 0.01$ | $0.93 \pm 0.01$ | $0.90 \pm 0.01$ | $0.90 \pm 0.01$ |
| 7 | $0.93 \pm 0.00$ | $0.86 \pm 0.01$ | $0.90 \pm 0.01$ | $0.90 \pm 0.01$ |
| 9 | $0.92 \pm 0.01$ | $0.91 \pm 0.01$ | $0.89 \pm 0.02$ | $0.89 \pm 0.02$ |
| | SSIM - Root dataset | | | |
| 3 | $0.86 \pm 0.01$ | $0.83 \pm 0.01$ | $0.84 \pm 0.02$ | $0.84 \pm 0.03$ |
| 5 | $0.85 \pm 0.01$ | $0.86 \pm 0.02$ | $0.84 \pm 0.02$ | $0.83 \pm 0.02$ |
| 7 | $0.84 \pm 0.01$ | $0.84 \pm 0.01$ | $0.84 \pm 0.02$ | $0.83 \pm 0.03$ |
| 9 | $0.84 \pm 0.01$ | $0.83 \pm 0.01$ | $0.85 \pm 0.01$ | $0.84 \pm 0.02$ |

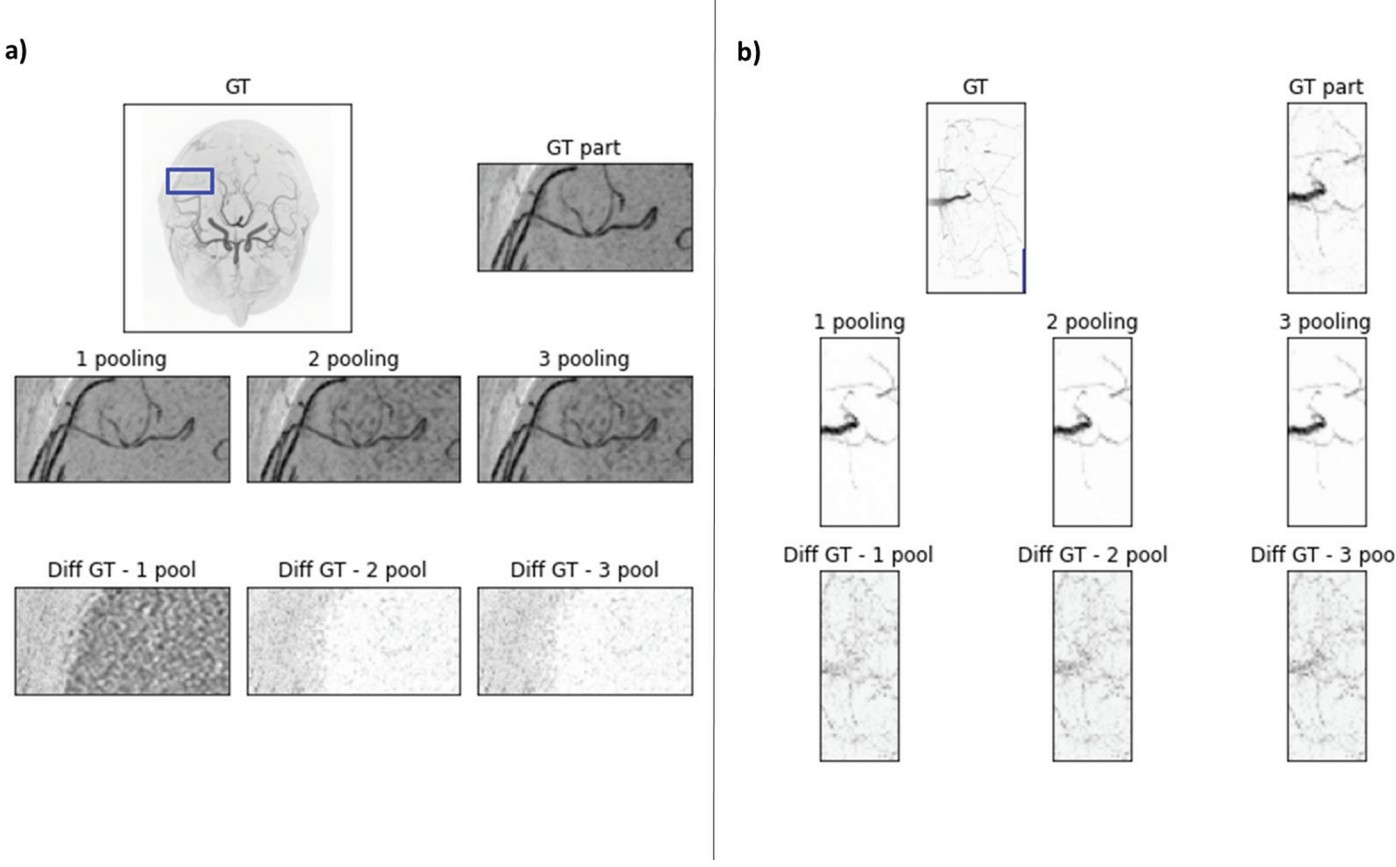

**Fig 4. Impact of pooling operations in the uPL on the results (Left: MIP of an example MRA image; and right: MR Root image. The marked region is the region which is displayed in large on the right.)** First row: GT image and zoomed part of GT image. Second row: Reconstructed image of image disturbed with 10% noise with different number of pooling operations in the uPL network. Third row: Difference image between GT and reconstructed image.

**Table 5. Mean and std SSIM values across different random seeds for both datasets calculated. Underlined are the values for the best loss function. I.e. if uPL outperformed L1-loss for a network architecture, the uPL values are underlined. All other evaluation metrics, std values across the testsets, and evaluation metrics calculated on image parts only are listed in the Supporting information (S6–S10 Tables).**

| Network/Loss | SSIM - MR root dataset | | | |
|---|---|---|---|---|
| | 1% noise | 5% noise | 10% noise | 20% noise |
| DnCNN/L1 | 0.83 ± 0.02 | 0.81 ± 0.01 | 0.79 ± 0.02 | 0.78 ± 0.01 |
| DnCNN/uPL | 0.86 ± 0.01 | 0.86 ± 0.01 | 0.86 ± 0.01 | 0.82 ± 0.0 |
| ResNet/L1 | 0.83 ± 0.02 | 0.83 ± 0.02 | 0.83 ± 0.02 | 0.80 ± 0.01 |
| ResNet/uPL | 0.86 ± 0.01 | 0.85 ± 0.01 | 0.84 ± 0.01 | 0.82 ± 0.0 |
| Transformer/L1 | 0.77 ± 0.02 | 0.79 ± 0.01 | 0.78 ± 0.02 | 0.72 ± 0.03 |
| Transformer/uPL | 0.81 ± 0.01 | 0.82 ± 0.01 | 0.82 ± 0.0 | 0.81 ± 0.01 |
| | SSIM - MRA dataset | | | |
| DnCNN/L1 | 0.86 ± 0.01 | 0.81 ± 0.01 | 0.79 ± 0.02 | 0.81 ± 0.01 |
| DnCNN/uPL | 0.92 ± 0.01 | 0.90 ± 0.01 | 0.87 ± 0.01 | 0.84 ± 0.02 |
| ResNet/L1 | 0.99 ± 0.01 | 0.87 ± 0.02 | 0.81 ± 0.01 | 0.84 ± 0.01 |
| ResNet/uPL | 0.98 ± 0.01 | 0.92 ± 0.01 | 0.85 ± 0.01 | 0.82 ± 0.02 |
| Transformer/L1 | 0.97 ± 0.03 | 0.91 ± 0.02 | 0.85 ± 0.02 | 0.68 ± 0.03 |
| Transformer/uPL | 0.98 ± 0.02 | 0.93 ± 0.01 | 0.83 ± 0.01 | 0.78 ± 0.02 |

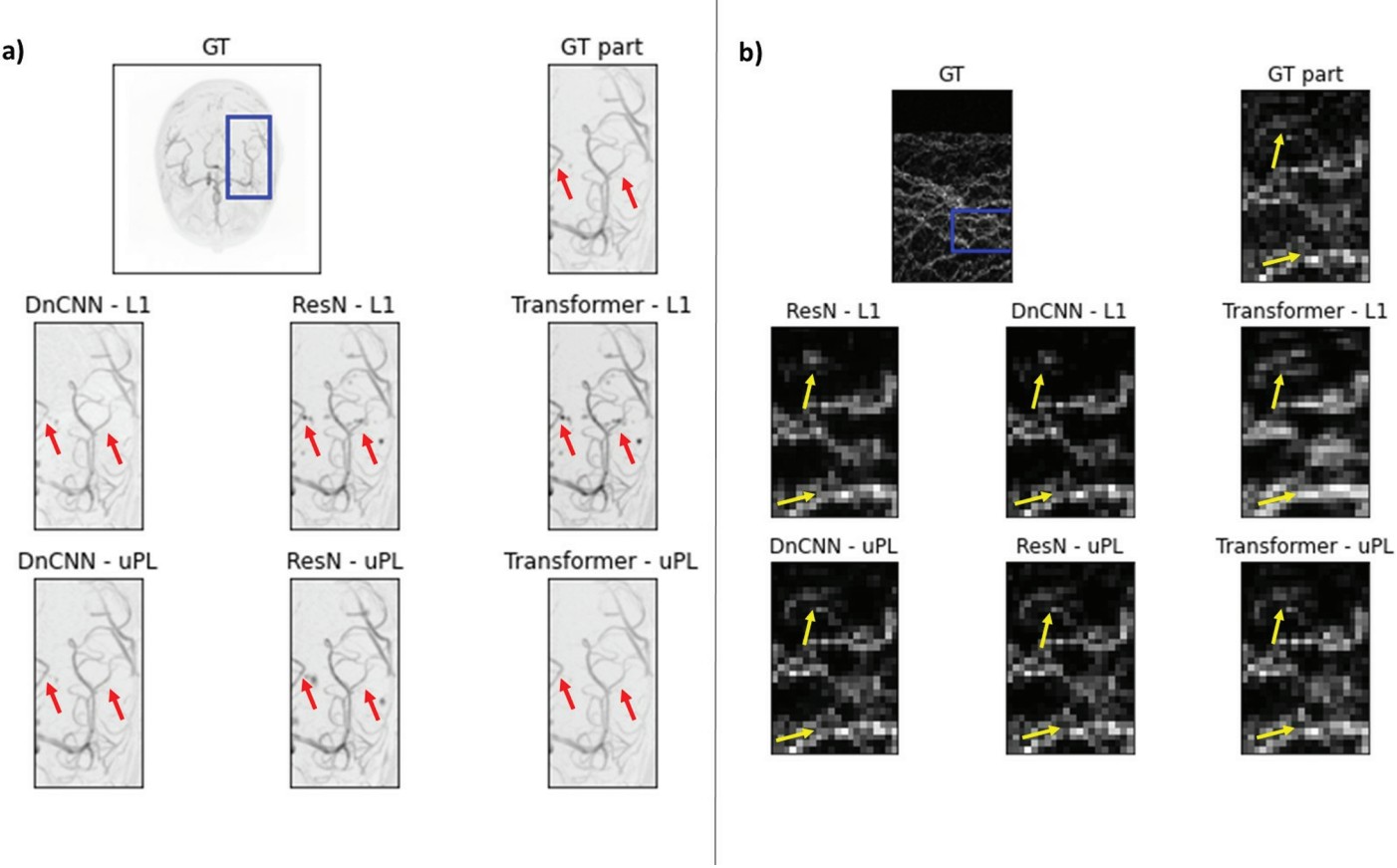

**Fig 5. MIP of original image (First row) and denoising results of image disturbed with 10% noise for different denoising networks and the L1-loss (second row) and our uPL.** Remarkable differences are marked with arrows.

(0.81 ± 0.01) for uPL and 0.68 ± 0.03 (0.72 ± 0.03) for L1 for the MRA (root) dataset and 20% noise). The benefit of uPL was similar for DnCNN and ResNet architectures.

As expected, denoising performance was best for the lowest noise levels: The uPL resulted in SSIM values of 0.82 - 0.86 for the roots data and of 0.9 - 0.99 for the MRA data. The L1 loss led to SSIM values between 0.81 - 0.84 for the root and 0.81 - 0.98 for the MRA data. The uPL outperformed especially in higher noise levels the L1 loss although the denoising performance decreases for higher noise levels. For the MRA data, the superiority in performance was not visible for the lowest and highest noise level and the ResNet, and 10% added noise and the Transformer network.

## Discussion

In summary, we demonstrated that the untrained Perceptual Loss leads to clear improvement for image denoising of MR images displaying line-like structures.

We observe that for both datasets, the uPL leads especially with small uPL networks to very good results. We demonstrated that loss network initialization has a minor impact on the results. However, more parameters might be important and future research may find better loss networks. For example, the use of other non-linearities than ReLU might influence the

results and introducing layer weights or masks may be pertinent (similar to the 'diversity loss' in [50]). We plan to address these points in future work.

It is hard to compare our results directly with results from other works due to the special characteristics of our datasets. Especially the root dataset is very sparse and yields very thin structures. As in our experiments, U-Net-like architectures failed for the MR root images, other famous denoising networks were not tested in this study [51,52]. We hypothesize that the down-sampling part of a U-Net architecture leads to a loss of information as the roots are very fine structures. We found only a small number of works that report on image denoising of MR angiographs. In [53] the authors used also an U-Net-shaped architecture. However, by testing three different denoising networks - one of them a Transformer network, we aimed to demonstrate that the success of the uPL is not network-dependent and can improve the results of different network architectures.

In this work, we focused on denoising of MR images displaying line-like structures. These images contain very fine details and yield therefore special challenges. In future work, we will investigate if the uPL can also be used for medical images yielding other image characteristics such as MR images of the human brain, PET or CT images.

Especially for difficult tasks, i.e. with increasing noise level the uPL performs considerably better than L1 loss for both datasets. This is probably because the uPL is taking neighborhood information into account by using convolutional kernels with sizes larger than 1 when calculating the loss. Therefore, even if one pixel is highly disturbed by noise, information about the ground truth pixel can still be estimated by its neighbors.

Regarding network architecture, we observed similar or even better performance when using uPL and a DnCNN than when combining L1 loss and one of the other network structures. These results indicate that the use of uPL makes it possible to use computationally less expensive networks without a performance drop. Additionally, our results suggest that the additional value of a Transformer network is not visible for both datasets. This is in contrast to other works that showed superior performance for similar denoising Transformers [54]. The lower performance of the Transformer network is likely due to the sparsity of the MR root and MRA images. Locally, both datasets contain a rather low variety of different features, i.e. 'just' lines and not rich texture. Therefore, large, computationally expensive, and complex networks such as the tested Transfomer variant may tend to overfit.

The success of perceptual loss was thought to be due to the features networks learned during training. However, as previous studies demonstrated for 2D images, pre-trained and untrained networks used in the loss function lead to comparable results [32,33]. This might be because randomly initialized weights can also represent the statistical properties of the training data. The pre-trained networks are usually trained on classification tasks. However, other image characteristics could be more important for image denoising of an entire image. E.g. the display of sharp edges can also be captured by convolutions and untrained feature maps. Additionally, the perceptual distance between two images is equivalent to the maximum mean discrepancy (MMD) distance between local distributions of small patches in the two images [55]. As also demonstrated in this study, different network structures in the loss function have a high impact on the denoising results. These findings suggest that the design of the network architecture used in the loss function can be chosen such that it captures the most important image information. With a best mean SSIM of 0.87 the denoising results for our root dataset still need improvement to allow for reduced measurement time in the targeted application. None of the tested denoising network/loss function combinations allow to recover very fine roots, even though the performance increase using our uPL is considerable. In consequence, as of now, plants containing very fine roots need to be scanned for a longer time than plants with thick roots. Therefore, before implementing the trained networks on a

plant root scanner, it needs to be verified for which plants (i.e. which root thickness) reduced image time in combination with a denoising network can be used without losing fine details.

Results are considerably better for the MRA dataset. With SSIM values between 0.99 and 0.93 for the lower noise levels, the proposed networks might be used for image denoising in a clinical setting. However, to apply the proposed approach in a clinical setting, our denoising framework needs to be verified on MRA data including healthy and unhealthy subjects (i.e. patients with e.g. an aneurysm). In a clinical scenario, it needs to be verified that denoising does not suppress important clinical information. This will be done in future studies based on this work.

Our study has several limitations. First, all networks were trained and applied on data acquired at the same scanner. In future work, we will train and test our approach on data from different scanners yielding different image characteristics. Second, the MRA dataset used in this study yields exclusively images from healthy subjects. In future work, we will apply the networks also to subjects with vessel abnormalities.

MR images of human subjects also suffer from motion artifacts. The proposed uPL might also be a promising candidate for training networks for MR motion correction what will be investigated in future work. All in all, our results demonstrate that the untrained Perceptual Loss can be used in image denoising networks. To what extent it can be used for image enhancement in a clinical scenario, needs to be determined in future work. However, the untrained perceptual loss should definitely considered when training neural networks on MR images displaying fine, line-like structures.

## Supporting information

**S1 Fig. ResNet architecture.** Illustration of the denoising network ResNet. The residual blocks are repeated five times. The first convolutional layer yields kernel size 9 with 64 output channels. All other convolutional layers yield kernel size 3 and 64 output channels. The last convolutional layer has kernel size 3 and 1 output channel.
(PDF)

**S2 Fig. Transformer architecture.** IIllustration of the denoising Transformer. The Transformer blocks are organized sequentially. Details about the Transformer blocks are explained in the corresponding paper for 2D.
(PDF)

**S3 Fig. DnCNN architecture.** Illustration of the denoising network DnCNN.
(PDF)

**S4 Fig. Illustration of evaluated image parts.** Illustration of cropped image parts on which evaluation metrics were calculated.
(PDF)

**S1 Table. Metrics Random Seeds.** Evaluation metrics for different random seeds for MRA and root dataset.
(PDF)

**S2 Table. Metrics different loss functions.** Evaluation metrics for MRA images and MR root images for the loss functions included in this study calculated on image parts only. Given are the mean and standard deviation values for the five random seeds used for network training.
(PDF)

**S3 Table. Mean and std values one random seed.** Mean and std values calculated over the testset for one random seed. Values are similar across random seeds.
(PDF)

**S4 Table. MSE for different kernel sizes/network depths.** MSE for different kernel sizes and network depth: MSE calculated only for roots regions (above) and MSE for the whole image also for the MR root dataset (below).
(PDF)

**S5 Table. Evaluation metrics for kernel sizes/network depths.** PSNR/MSE for different kernel sizes and network depth for MRA (above) and MSE for the MRA dataset (below).
(PDF)

**S6 Table. PSNR values for different network architectures.** PSNR values for both datasets, network structures, and noise levels.
(PDF)

**S7 Table. MSE values calculated on image parts for different network architectures.** MSE values only calculated on the roots part and MSE values for both datasets for all network architectures, and noise levels included in this study.
(PDF)

**S8 Table. SSIM values calculated on image parts for different network architectures.** SSIM values for both datasets calculated on the center of the image.
(PDF)

**S9 Table. PSNR values calculated on image parts for different network architectures.** PSNR values for both datasets calculated on the center of the image.
(PDF)

**S10 Table. MSE values calculated on image parts for different network architectures.** MSE values for both datasets calculated on the center of the image.
(PDF)

## Acknowledgments

We acknowledge the help of Dagmar van Dusschoten in providing MR images. The authors gratefully acknowledge the computing time granted through JARA on the supercomputer JURECA [56] at Forschungszentrum Jülich.

## Author contributions

**Conceptualization:** Elisabeth Pfaehler, Daniel Pflugfelder, Hanno Scharr.

**Data curation:** Daniel Pflugfelder.

**Formal analysis:** Elisabeth Pfaehler.

**Funding acquisition:** Elisabeth Pfaehler.

**Methodology:** Elisabeth Pfaehler.

**Project administration:** Hanno Scharr.

**Resources:** Daniel Pflugfelder.

**Software:** Elisabeth Pfaehler.

**Supervision:** Hanno Scharr.

**Writing – original draft:** Elisabeth Pfaehler.

**Writing – review & editing:** Elisabeth Pfaehler, Daniel Pflugfelder, Hanno Scharr.

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
