## [Decision Letter · Decision Letter 0]

9 Sep 2024

PONE-D-24-26390

Untrained Perceptual Loss for image denoising of line-like structures in MR images

PLOS ONE

Dear Dr. Pfaehler,

Thank you for submitting your manuscript to PLOS ONE. After careful consideration, we feel that it has merit but does not fully meet PLOS ONE’s publication criteria as it currently stands. Therefore, we invite you to submit a revised version of the manuscript that addresses the points raised during the review process.

We look forward to receiving your revised manuscript.

Kind regards,

Khan Bahadar Khan, Ph.D

Academic Editor

PLOS ONE

Journal Requirements:

"This work was supported by the Networking Funds of the Helmholtz Association of German Research Centres 276 HighLine ZT-I-PF-4-042)."

3. Please note that funding information should not appear in the Acknowledgments section or other areas of your manuscript. We will only publish funding information present in the Funding Statement section of the online submission form. Please remove any funding-related text from the manuscript. 

6. Please ensure that you refer to Figure 1 in your text as, if accepted, production will need this reference to link the reader to the figure.

7. We notice that your supplementary figures and tables are included in the manuscript file. Please remove them and upload them with the file type 'Supporting Information'. Please ensure that each Supporting Information file has a legend listed in the manuscript after the references list.

Reviewers' comments:

Reviewer's Responses to Questions

**Comments to the Author**

1. Is the manuscript technically sound, and do the data support the conclusions?

Reviewer #1: Yes

Reviewer #2: Yes

2. Has the statistical analysis been performed appropriately and rigorously? 

Reviewer #1: Yes

Reviewer #2: No

3. Have the authors made all data underlying the findings in their manuscript fully available?

Reviewer #1: Yes

Reviewer #2: No

4. Is the manuscript presented in an intelligible fashion and written in standard English?

Reviewer #1: Yes

Reviewer #2: Yes

5. Review Comments to the Author

Reviewer #1: This study investigated the effect of using a perceptual loss function with an untrained network on 3D MRI denoising.

The authors tested various untrained networks with different initializations and structures, including depth, kernel size, and pooling operations. Since obtaining a trained network to calculate perceptual loss is often challenging, the use of a perceptual loss function with an untrained network appears to have significant utility. The paper is clearly written, and appropriate results are presented to support each claim. However, further analysis of the untrained perceptual loss is needed in addition to the evaluation results. There are also a few additional contents that should be included in the paper.

1. The evaluation metrics used in this paper are predominantly influenced by large structures in the image. However, the regions that need to be restored through denoising are the fine structures. Therefore, the evaluation results for the fine structure areas (for example, the left half of the image in Fig. 2) should be presented alongside the evaluation results for the entire image.

2. Line 254: uPL takes neighborhood information into account when calculating loss. What is the basis for this claim?

3. It is thought that the effectiveness of perceptual loss arises because the trained network extracts meaningful feature maps. There have been papers that trained a network specifically to extract meaningful features for the primary task, and these papers indeed improved the performance of the primary task. Therefore, a discussion is needed on why an untrained network is also effective.

Reviewer #2: The manuscript describes a new deep learning approach to the denoising of magnetic resonance (MR) images, especially those depicting fine fiber structures. The core idea is to use an “untrained perceptual loss” (uPL) to drive the training of the deep neural networks (DNN) intended for denoising. This approach is not new, but here applied and benchmarked systematically for 3D MR image stacks for the first time. The results show that the uPL loss is actually superior to other commonly used loss functions in this domain, at least for the chosen datasets and denoising network architectures.

Overall assessment

The scientific idea and results are definitely worth publishing because they provide some new unique insights. However, the writing needs some improvement (more on this in the next paragraph), and the presented experiments seem to be carried out only once for every task condition. Thus, the results which we find in the tables are derived from a single training run of a single network. We all know that DNN training is a stochastic process where the outcome can vary from training run to training run. Therefore, I strongly recommend to carry out multiple training runs per task condition and to present the mean outcome in the tables (incl. standard deviation).

Even with the current data, one should report for every value in the result tables the standard deviations related to the variability in the test set and the number of test samples. In this way it would be possible to estimate if any of the observed differences are significant in a statistical sense. Even better would be proper statistical testing by the authors to convince the reader that the reported differences in the various performance measures are meaningful at all. For now, in my current assessment of the paper, I *assume* that this is the case for most reported differences in SSIM, PSNR, and MSE values.

The abstract and introduction section are written very well. For the first paragraph of the introduction, the following reference may be a good addition: ZH Shah, M Müller, B Hammer, T Huser, and W Schenck. Impact of different loss functions on denoising of microscopic images. In 2022 International Joint Conference on Neural Networks (IJCNN), 2022.

(in this paper, various loss functions for denoising are compared with each other, among them a version of the perceptual loss; to include this reference is just a suggestion because it seems to be a good fit, but it is not a must)

The section on “Materials and Methods” lacks some clarity, however. Starting at the subsection “Conventional loss functions”, the reader gets easily confused how the different uPL variations are related to each other. First, VGG19, AlexNet and a simple CNN are mentioned. It is not explicitly written that only the simple CNN serves as starting point for the variations of weight initialization, depth, and number of pooling layers. It is also not clear from the writing how the latter three variations interact with each other.

Regarding the training details, the question is why 30.000 iterations were chosen. Is this the optimum number for the performance measures on the validation sets? No over- or underfitting in any of the experimental variations?

Also the learning rate of 0.001 needs justification.

The writing in the “Results” section should also be improved because it is partly confusing. First, I strongly recommend to check and improve grammar and sentence structure, second, to state in the beginning of each subsection what is the goal of the described experiment and what is the experimental configuration (because it does not always fully align with the corresponding part in the “Methods” section).

Also reg. the “Results” section, I recommend to add more visualizations of the denoising results. The existing figures and possible additional figures could be formatted in a nicer and more compact way.

The “Discussion” section is appropriate.

Small remarks

• writing style not consistent reg. “2D” vs. “2d” and “3D” vs. “3d”

• writing style not consistent reg. “transformer” vs. “Transformer”

• p. 4, beginning: No figure for DuCNN in suppl. materials

• p. 4, 3rd paragraph: More in-depth explanation about adaption of transformer architecture to 3D setting would be nice

• p. 6, line 190-192: If uniform init. is worse than Xavier normal, why use default init. in “all other experiments”?

• line 195: “slighlty”  “slightly”

• caption of table 3: “… MRA (above) for…”  “… MRA (above) and for…”

• line 201: “…show the number…”  “…show that the number…”

• line 204-208: Description of the results for MRA does not reflect the similarly good results for 7 conv. layers

• caption of Fig. 3 reg. “difference images”: Difference to what?

• line 226/227: I don’t understand this sentence since the results in table 4 show consistently better results for uPL at the highest noise level with the roots dataset.

• Caption of table 4: There are no other eval. metrics listed in the suppl. materials section reg. table 4 (btw, what is “Section 7”?)

• Figures S2 and S3: Not very helpful for better understanding (esp. S2)

• Figure S4: Subfigure (g) obviously missing

• Caption of Table S2: Very confusing

• Caption of Table S3: Replace “SSIM” with “PSNR/MSE”

6. PLOS authors have the option to publish the peer review history of their article (what does this mean?). If published, this will include your full peer review and any attached files.

Reviewer #1: No

Reviewer #2: No

---

## [Author Response · Author response to Decision Letter 1]

16 Oct 2024

We thank the reviewers for their valuable feedback which helped to increase the quality of the manuscript. We adapted the manuscript accordingly. Our detailed answers are listed below.

Reviewer #1:

1. The evaluation metrics used in this paper are predominantly influenced by large structures in the image. However, the regions that need to be restored through denoising are the fine structures. Therefore, the evaluation results for the fine structure areas (for example, the left half of the image in Fig. 2) should be presented alongside the evaluation results for the entire image.

We thank the reviewer for this important point. We now also calculated the image quality metrics for the cropped region of the images. All these regions also contained vessels/roots (i.e. the important parts of the images), we calculated the image quality metrics also on these regions. To avoid confusion, we added the results to the supplemental information. We added the following sentence to the manuscript:

‚As both datasets contain fine structures, the evaluation metrics were additionally calculated on the parts of the image containing important information. A cube of size 52 x 52 x 52 was cropped from the root data. As the roots grow from up to down, the cube was cropped from the upper middle part of the image. A cube of size 68 x 68 x 68 was cropped from the middle of the MRA images as these images contain important information mainly in the image center. An example illustration is displayed in Supplemental Figure S2 . These results are reported in the Supplemental Information.‘

We observed different, but consistent results when calculating evaluation metrics on the whole image and on image parts only (i.e. for the root dataset, 10% noise, and DnCNN network: L1 Loss SSIM: 0.73, uPL: SSIM 0.77 when calculated on image parts; fort he MRA dataset, 10% noise and DnCNN network: L1: SSIM 0.83, uPL: 0.87)

2. Line 254: uPL takes neighborhood information into account when calculating loss. What is the basis for this claim?

As the uPL compares feature maps that are acquired by applying convolutional kernels with a kernel size > 1, these feature maps also contain information about the voxel neighborhood. We added the following sentence to the manuscript for clarification: ‚This is probably because the uPL is taking neighborhood information into account by using convolutional kernels with sizes larger than 1 when calculating the loss.‘

3. It is thought that the effectiveness of perceptual loss arises because the trained network extracts meaningful feature maps. There have been papers that trained a network specifically to extract meaningful features for the primary task, and these papers indeed improved the performance of the primary task. Therefore, a discussion is needed on why an untrained network is also effective.

Thank you for this question. We added a part to the discussion responding this question. It reads: ‚ The success of perceptual loss was thought to be due to the features networks learned during training. However, as previous studies demonstrated for 2D images, pre-trained and untrained networks used in the loss function lead to comparable results (Liu et al., Generic perceptual loss for modeling structured output dependencies, Proceedings of the IEEE/CVF Conference on Computer Vision and Pattern Recognition). This might be because randomly initialized weights can also represent the statistical properties of the training data. The pre-trained networks are usually trained on classification tasks. However, other image characteristics could be more important for image denoising of an entire image. E.g. the display of sharp edges can also be captured by convolutions and untrained feature maps. Additionally, the perceptual distance between two images is equivalent to the maximum mean discrepancy (MMD) distance between local distributions of small patches in the two images (Amir and Weiss, Understanding and Simplifying Perceptual Distances, CVPR 2021). As also demonstrated in this study, different network structures in the loss function have a high impact on the denoising results. These findings suggest that the design of the network architecture used in the loss function can be chosen such that it captures the most important image information. ‘

Reviewer #2: The manuscript describes a new deep learning approach to the denoising of magnetic resonance (MR) images, especially those depicting fine fiber structures. The core idea is to use an “untrained perceptual loss” (uPL) to drive the training of the deep neural networks (DNN) intended for denoising. This approach is not new, but here applied and benchmarked systematically for 3D MR image stacks for the first time. The results show that the uPL loss is actually superior to other commonly used loss functions in this domain, at least for the chosen datasets and denoising network architectures.

Overall assessment

The scientific idea and results are definitely worth publishing because they provide some new unique insights. However, the writing needs some improvement (more on this in the next paragraph), and the presented experiments seem to be carried out only once for every task condition. Thus, the results which we find in the tables are derived from a single training run of a single network. We all know that DNN training is a stochastic process where the outcome can vary from training run to training run. Therefore, I strongly recommend to carry out multiple training runs per task condition and to present the mean outcome in the tables (incl. standard deviation).

We thank the reviewer for pointing this out. We now ran the networks for each condition five times (with five different random seeds). We report the mean and std values across random seeds in the Tables. We added the following sentence to the Materials and Methods section: ‚All experiments were performed with five different random seeds.‘

Even with the current data, one should report for every value in the result tables the standard deviations related to the variability in the test set and the number of test samples. In this way it would be possible to estimate if any of the observed differences are significant in a statistical sense. Even better would be proper statistical testing by the authors to convince the reader that the reported differences in the various performance measures are meaningful at all. For now, in my current assessment of the paper, I *assume* that this is the case for most reported differences in SSIM, PSNR, and MSE values.

We thank the reviewer for pointing this out; We added the standard deviation across random seeds to the results. The results for one random seed including the standard deviation in the testset is separately displayed in the supplemental material. However, we hope and believe with the application of five random seeds and the additional evaluation on image parts only (requested by Reviewer 1), our results are convincing.

The abstract and introduction section are written very well. For the first paragraph of the introduction, the following reference may be a good addition: ZH Shah, M Müller, B Hammer, T Huser, and W Schenck. Impact of different loss functions on denoising of microscopic images. In 2022 International Joint Conference on Neural Networks (IJCNN), 2022.

(in this paper, various loss functions for denoising are compared with each other, among them a version of the perceptual loss; to include this reference is just a suggestion because it seems to be a good fit, but it is not a must)

We thank the reviewer for mentioning this paper which is indeed a very good fit. We added it to the introduction section: ‚The Perceptual Loss is also often used in combination with L1- or L2-loss where it also leads to performance improvements ‘

The section on “Materials and Methods” lacks some clarity, however. Starting at the subsection “Conventional loss functions”, the reader gets easily confused how the different uPL variations are related to each other. First, VGG19, AlexNet and a simple CNN are mentioned. It is not explicitly written that only the simple CNN serves as starting point for the variations of weight initialization, depth, and number of pooling layers. It is also not clear from the writing how the latter three variations interact with each other.

We adapted ‚Materials and Methods‘ section accordingly. We changed the order of the subsections: We now first introduce the untrained perceptual loss and then introduce a simple ‚proof of concept‘ study where we compare the performance of a simple uPL network with the performance of other networks. We then elaborate on the impact of different uPL network characteristics on the results.

Regarding the training details, the question is why 30.000 iterations were chosen. Is this the optimum number for the performance measures on the validation sets? No over- or underfitting in any of the experimental variations?

Also the learning rate of 0.001 needs justification.

We experimented with different learning rates and numbers of iterations. The chosen settings led to the overall optimal performance in the validation sets. As performance was similar in the training set, we assume no over- or underfitting occured for this numbers. We added the following part to the manuscript:

‚The training parameters were chosen as they lead to the overall best performance in the validation set.‘

The writing in the “Results” section should also be improved because it is partly confusing. First, I strongly recommend to check and improve grammar and sentence structure, second, to state in the beginning of each subsection what is the goal of the described experiment and what is the experimental configuration (because it does not always fully align with the corresponding part in the “Methods” section).

We thank the reviewer for this important point. We changed the wording and structure of this section accordingly. We made some small changes in each part and added a sentence describing the aim of each section.

Also reg. the “Results” section, I recommend to add more visualizations of the denoising results. The existing figures and possible additional figures could be formatted in a nicer and more compact way.

We changed the Figures and added some results.

The “Discussion” section is appropriate.

Small remarks

• writing style not consistent reg. “2D” vs. “2d” and “3D” vs. “3d”

Thank you for pointing this out. We changed all namings now to 2D and 3D.

• writing style not consistent reg. “transformer” vs. “Transformer”

Done

• p. 4, beginning: No figure for DuCNN in suppl. Materials

Done

• p. 4, 3rd paragraph: More in-depth explanation about adaption of transformer architecture to 3D setting would be nice

We added a more detailed explanation. Additionally, we changed the Supplemental Figure and added more details. The section reads now:

The Transformer network is specially designed such that it can be used for large images while modeling global connectivity. In this network, multi-head ’transposed’ attention (MDTA) blocks are introduced applying attention across feature dimensions rather than across spatial dimensions. Before feeding the data to the MDTA blocks, the images are resized to the size (image height · image width) × number of channels. After the MDTA blocks follow feed-forward blocks consisting of two convolutional layers and a gating layer. We adjust the transposition in the attention blocks to the 3D case. In the 3D Transformer network, all 2D convolutions are replaced by 3D convolutions. Before the data is fed to the MDTA blocks, the data is resized to the size (image height· image width · image depth) × number of channels. As for our fine structures, UNet-like architectures lead to performance drops, the Transformer blocks were in our study combined sequentially. A graphical overview of all network structures is displayed in the supplemental material (S1 Fig, S2 Fig, S3 Fig).

• p. 6, line 190-192: If uniform init. is worse than Xavier normal, why use default init. in “all other experiments”?

We apologize, this was a mistake. We trained all networks with xavier normal initialisation. We changed the sentence in the manuscript accordingly.

• line 195: “slighlty”  “slightly”

Done.

• caption of table 3: “… MRA (above) for…”  “… MRA (above) and for…”

Done

• line 201: “…show the number…”  “…show that the number…”

Done

• line 204-208: Description of the results for MRA does not reflect the similarly good results for 7 conv. Layers

Done

• caption of Fig. 3 reg. “difference images”: Difference to what?

TODO

• line 226/227: I don’t understand this sentence since the results in table 4 show consistently better results for uPL at the highest noise level with the roots dataset.

True. We changed the sentence accordingly to: For the MRA data, the superiority in performance was not visible for the lowest and highest noise level and the ResNet, and 10 % added noise and the Transformer network.

• Caption of table 4: There are no other eval. metrics listed in the suppl. materials section reg. table 4 (btw, what is “Section 7”?)

Apologize for this mistake. We added the tables to the Supplemental information.

• Figures S2 and S3: Not very helpful for better understanding (esp. S2)

We changed the corresponding figures accordingly.

• Figure S4: Subfigure (g) obviously missing

Done

• Caption of Table S2: Very confusing

We changed the caption to: MSE for different kernel sizes and network depth: MSE calculated only for roots regions (above) and MSE for the whole image also for the MR root dataset (below)

• Caption of Table S3: Replace “SSIM” with “PSNR/MSE”

Done.

---

## [Decision Letter · Decision Letter 1]

27 Nov 2024

PONE-D-24-26390R1

Untrained Perceptual Loss for image denoising of line-like structures in MR images

PLOS ONE

Dear Dr. Pfaehler,

Thank you for submitting your manuscript to PLOS ONE. After careful consideration, we feel that it has merit but does not fully meet PLOS ONE’s publication criteria as it currently stands. Therefore, we invite you to submit a revised version of the manuscript that addresses the points raised during the review process.

We look forward to receiving your revised manuscript.

Kind regards,

Khan Bahadar Khan, Ph.D

Academic Editor

PLOS ONE

Reviewers' comments:

Reviewer's Responses to Questions

**Comments to the Author**

1. If the authors have adequately addressed your comments raised in a previous round of review and you feel that this manuscript is now acceptable for publication, you may indicate that here to bypass the “Comments to the Author” section, enter your conflict of interest statement in the “Confidential to Editor” section, and submit your "Accept" recommendation.

Reviewer #1: All comments have been addressed

Reviewer #3: (No Response)

Reviewer #4: (No Response)

2. Is the manuscript technically sound, and do the data support the conclusions?

Reviewer #1: Yes

Reviewer #3: Yes

Reviewer #4: (No Response)

3. Has the statistical analysis been performed appropriately and rigorously? 

Reviewer #1: Yes

Reviewer #3: N/A

Reviewer #4: (No Response)

4. Have the authors made all data underlying the findings in their manuscript fully available?

Reviewer #1: No

Reviewer #3: Yes

Reviewer #4: (No Response)

5. Is the manuscript presented in an intelligible fashion and written in standard English?

Reviewer #1: Yes

Reviewer #3: Yes

Reviewer #4: (No Response)

6. Review Comments to the Author

Reviewer #1: This study investigated the effect of using a perceptual loss function with an untrained network on 3D MRI denoising.

The authors tested various untrained networks with different initializations and structures, including depth, kernel size, and pooling operations. Since obtaining a trained network to calculate perceptual loss is often challenging, the use of a perceptual loss function with an untrained network appears to have significant utility. The paper is clearly written, and appropriate results are presented to support each claim.

The authors have satisfactorily addressed my comments.

Reviewer #3: This is an interesting paper describing the use of uPL in conjunction with deep neural networks for denoising MR images to resolve fine structures. The paper is well written and provides useful information on the effectiveness of the presented technique.

General:

I think the figure and table captions need revision. They are not very clear and are missing information.

The discussion could be revised. It is a bit anemic. I would include a brief discussion of the limitations of your approach. I would also consider mentioning the MRA results, since you mention the root results.

What is the practical application implication of these results? You mention reduced imaging time, but to what extent would this allow that? You also mention the ability to use less computationally expensive denoising, but there is not an investigation of comparative performance in that regard, so it feels out-of-context. What specific advantage is gained from using this approach?

Comments:

Lines 70-72 “For both datasets, four levels of Rician noise were artificially added to assess the impact of the loss functions for different signal-to-noise ratios (1%, 5%, 10%, and 20% noise added)”

I would be interested to see images with the various noise levels. It may be helpful for contextualizing the quality of denoising. I do not see these in your figures containing the GT images and denoised images.

Line 181 “The training parameters were chosen as they lead to the overall best performance in the validation sets”

Could you expand a bit on what you mean by validation sets? What exactly did you do to determine that these were the best parameters? I don’t think you need a lot of extra detail here, but it would be nice to understand how you came to these training parameters, since these are fairly important for the overall performance of the network and since you are investigating other parameters in depth.

Lines 190-192 "A cube of size 52 × 52 × 52 was cropped from the root data. 190 As the roots grow from top to bottom, the cube is cropped from the upper middle part of the image. A cube of size 68 × 68 × 68 is cropped from the middle of the MRA"

Is there a reason for these cube sizes? What does middle and upper middle mean in this context? Where all sample cubes taken from the same coordinates?

Figures 2, 3, 4

I think these figures could use better captions – they are missing some key info. For example, in Fig 2, you do not mention the noise level, or what exactly is being displayed in the caption (but then describe it in text). This should be moved to the caption for easier reference. The same holds true for figures 2 & 3. I see that this info is included elsewhere, but as a reader, it is difficult to keep track of what is being displayed.

Figure 4

There are arrows (presumably pointing to features of interest) but they are not explained. Why have you chosen to highlight these points?

Tables

Why are some values underlined? It would be helpful to provide a legend/explanation in the caption for this. In general, I think the table and figure captions could be revised for clarity and content.

Line 253 “Denoising network architecture vs. loss function for different noise levels”

Why only use L1 here as a comparison?

Lines 328-334

This section of text feels somewhat out of place.

Lines 335-336 “However, our results demonstrate the benefit of the untrained Perceptual Loss for both 3D datasets and all noise levels.”

This final sentence should be reworked a bit. It doesn’t flow particularly well with the prior paragraph, and I think that a final statement/claim like this should be given more attention and justification.

Reviewer #4: (No Response)

7. PLOS authors have the option to publish the peer review history of their article (what does this mean?). If published, this will include your full peer review and any attached files.

Reviewer #1: No

Reviewer #3: No

Reviewer #4: No

---

## [Author Response · Author response to Decision Letter 2]

16 Jan 2025

We thank the reviewers for their valuable time and feedback which helped to improve the manuscript. Detailed answers are given below.

Reviewer 3:

I think the figure and table captions need revision. They are not very clear and are missing information.

The discussion could be revised. It is a bit anemic. I would include a brief discussion of the limitations of your approach. I would also consider mentioning the MRA results, since you mention the root results.

We thank the reviewer for this comment. We added the following part to the discussion: ‚Results are considerably better for the MRA dataset. With SSIM values between 0.99 and 0.93 for the lower noise levels, the proposed networks might be used for image denoising in a clinical setting. However, to apply the proposed approach in a clinical setting, our denoising framework needs to be verified on MRA data including healthy and unhealthy subjects (i.e. patients with e.g. an aneurysm). In a clinical scenario, it needs to be verified that denoising does not suppress important clinical information. This will be done in future studies based on this work.’

Moreover, we included a paragraph about the limitations of our study. ‚Our study has several limitations. First, all networks were trained and applied on data acquired at the same scanner. In future work, we will train and test our approach on data from different scanners yielding different image characteristics. Second, the MRA dataset used in this study yields exclusively images from healthy subjects. In future work, we will apply the networks also to subjects with vessel abnormalities.

 MR images of human subjects also suffer from motion artifacts. The proposed uPL might also be a promising candidate for training networks for MR motion correction what will be investigated in future work.’

What is the practical application implication of these results? You mention reduced imaging time, but to what extent would this allow that? You also mention the ability to use less computationally expensive denoising, but there is not an investigation of comparative performance in that regard, so it feels out-of-context. What specific advantage is gained from using this approach?

The proposed approach can be used for eliminating noise from MR images containing line-like structures. This would e.g. allow shorter scan times. However, to what extent this can be used in the clinic or for plant research, still needs to be investigated. For the use in a clinical setting, the network needs to be applied on a variety of images including data from healthy and sick subjects. Likely, also additional methods indicating e.g. the network uncertainty (i.e. ensemble models) need to be employed before a clinical implementation can be realized. For the use in plant science, it needs to be very carefully investigated for which kind of plants the method can be used. However, these investigations will be performed in future work. The proposed approach demonstrates that the uPL is a promising candidate for MR image denoising of line-like structures and should definitely be considered in a clinical application. We mention this limitation for root data already in the discussion: ‚With a best mean SSIM of 0.87 the denoising results for our root dataset still need improvement to allow for reduced measurement time in the targeted application.‘ And added a part regarding the MRA dataset: ‚However, to apply the proposed approach in a clinical setting, our denoising framework needs to be verified on MRA data including healthy and unhealthy subjects (i.e. patients with e.g. an aneurysm). In a clinical scenario, it needs to be verified that denoising does not suppress important clinical information. This will be done in future studies based on this work.‘

Comments:

Lines 70-72 “For both datasets, four levels of Rician noise were artificially added to assess the impact of the loss functions for different signal-to-noise ratios (1%, 5%, 10%, and 20% noise added)”

I would be interested to see images with the various noise levels. It may be helpful for contextualizing the quality of denoising. I do not see these in your figures containing the GT images and denoised images.

Thank you for this remark. We added a new Figure (Figure 2) displaying an example root and MRA image with different noise levels.

Line 181 “The training parameters were chosen as they lead to the overall best performance in the validation sets”

Could you expand a bit on what you mean by validation sets? What exactly did you do to determine that these were the best parameters? I don’t think you need a lot of extra detail here, but it would be nice to understand how you came to these training parameters, since these are fairly important for the overall performance of the network and since you are investigating other parameters in depth.

We thank the reviewer for this comment. As described in section ‘Datasets’, we split datasets in 3 parts, a training, a validation, and a test set. As usual in machine learning experimentation, the validation set was used for validation purposes during training, allowing e.g. for parameter tuning. Tests were run on the test set. We added the following paragraph to the manuscript: ‚For this purpose, networks were trained for 10.000, 15.000, 20.000, ... 50.000 iterations, batch size was set to 4, 8, 16, 24, and 32, and the learning rate was set to 0.001,0.002, ... 0.005. All possible combinations of these parameters were investigated and the parameters leading to the best performance in the above defined validation datasets were selected.’

Lines 190-192 "A cube of size 52 × 52 × 52 was cropped from the root data. 190 As the roots grow from top to bottom, the cube is cropped from the upper middle part of the image. A cube of size 68 × 68 × 68 is cropped from the middle of the MRA"

Is there a reason for these cube sizes? What does middle and upper middle mean in this context? Where all sample cubes taken from the same coordinates?

We thank the reviewer for this comment. The objective of calculating the evaluation metrics on a part from the image was to calculate the metrics on regions only containing important information and no background. Therefore, we chose the cube size such that as little background as possible was present in the image part. We also tested with slightly varying cube sizes and did not observe notable changes. For the MRA images, the cropped part does not contain any background and by placing the cube on the location of the center of the root, as little background as possible is included. As the MRA images contain different numbers of z-slices, the cube was in these cases not placed at the same coordinates. For the MR-root images, the same coordinates were always chosen. We changed the text to make this more clear: ‚A cube of size 52×52×52 was cropped from the root data. As the roots grow from top to bottom and because the plant is always placed in the center of the scanner, the image was cropped such that the upper, middle image part was always present in the evaluation. I.e. the image was cropped from z-slices 0 - 52, and x-, and y-slices from 70-132. A cube of size 68×68×68 is cropped from the middle of the MRA images as these images contain important information mainly in the image center i.e. the center of the image was determined and a cube with the mentioned size was cropped such that the center of the cube aligned with the center of the image.‘

Figures 2, 3, 4

I think these figures could use better captions – they are missing some key info. For example, in Fig 2, you do not mention the noise level, or what exactly is being displayed in the caption (but then describe it in text). This should be moved to the caption for easier reference. The same holds true for figures 2 & 3. I see that this info is included elsewhere, but as a reader, it is difficult to keep track of what is being displayed.

We thank the reviewer for this comment. We changed all captions accordingly.

Figure 4

There are arrows (presumably pointing to features of interest) but they are not explained. Why have you chosen to highlight these points?

Thank you for this remark. We highlighted these points to mark differences across images. We added this to the Figure captions.

Tables

Why are some values underlined? It would be helpful to provide a legend/explanation in the caption for this. In general, I think the table and figure captions could be revised for clarity and content.

We thank the reviewer for pointing this out. We changed the table captions to improve clarity.

Line 253 “Denoising network architecture vs. loss function for different noise levels”

Why only use L1 here as a comparison?

As we investigated a large number of parameters and network architectures, we decided to only include the results of the L1-loss to keep a better overview. The general trend that the uPL loss outperforms other loss networks is already shown in the previous experiments and would not add valuable information here.

Lines 328-334

This section of text feels somewhat out of place.

We changed the text to: ‚None of the tested denoising network/loss function combinations allow to recover very fine roots, even though the performance increase using our uPL is considerable. In consequence, as of now, plants containing very fine roots need to be scanned for a longer time than plants with thick roots. Therefore, before implementing the trained networks on a plant root scanner, it needs to be verified for which plants (i.e. which root thickness) reduced image time in combination with a denoising network can be used without losing fine details.’

Lines 335-336 “However, our results demonstrate the benefit of the untrained Perceptual Loss for both 3D datasets and all noise levels.”

This final sentence should be reworked a bit. It doesn’t flow particularly well with the prior paragraph, and I think that a final statement/claim like this should be given more attention and justification.

Thank you for this comment. We changed the text to: ‚All in all, our results demonstrate that the untrained Perceptual Loss can be used in image denoising networks. To what extent it can be used for image enhancement in a clinical scenario, needs to be determined in future work. However, the untrained perceptual loss should definitely considered when training neural networks on MR images displaying fine, line-like structures.’

Reviewer 4:

The topic is relevant and may be of interest to a broad range of the journal's readers. However, this reviewer has some major concerns about the paper.

The abstract does not highlight the specifics of the research or findings but contains too much background information. It is good to provide some specifics (e.g., sample size, dataset size, numbers from results, etc.).

We thank the reviewer for pointing this out. We changed the abstract accordingly. We added the following parts: ‚In this work, we concentrate on image denoising of MR images containing line-like structures such as roots or vessels. In particular, we investigate if the special characteristics of these datasets (connectivity, sparsity) benefit from the use of special loss functions for network training.’; ‚In this study, 536 MR images of plant roots in soil and 450 MRA images are included. The plant root dataset is split to 380, 80, and 76 images for training, validation, and testing. The MRA dataset is split to 300, 50, and 100 images for training, validation, and testing.’; ‚Our results are compared with the frequently used L1 loss for different network architectures. We observe, that our uPL outperforms conventional loss functions such as the L1 loss or a loss based on the Structural Similarity Index Metric (SSIM). For MRA images the uPL leads to SSIM values of 0.93 while L1 and SSIM loss led to SSIM values of 0.81 and 0.88, respectively. The uPL network’s initialization is not important (e.g. for MR root images SSIM differences of 0.01 occur across initializations, while network depth and pooling operations impact denoising performance slightly more (SSIM of 0.83 for 5 convolutional layers and kernel size 3 vs. 0.86 for 5 convolutional layers and kernel size 5 for the root dataset).‘

2) The discussion of the results does not highlight the strengths and weaknesses of the proposed approach.

Thank you for pointing this out. We changed the discussion accordingly and added the following part: ‚None of the tested denoising network/loss function combinations allow to recover very fine roots, even though the performance increase using our uPL is considerable. In consequence, as of now, plants containing very fine roots need to be scanned for a longer time than plants with thick roots. Therefore, before implementing the trained networks on a plant root scanner, it needs to be verified for which plants (i.e. which root thickness) reduced image time in combination with a denoising network can be used without losing fine details. Results are considerably better for the MRA dataset. With SSIM values between 0.99 and 0.93 for the lower noise levels, the proposed networks might be used for image denoising in a clinical setting. However, to apply the proposed approach in a clinical setting, our denoising framework needs to be verified on MRA data including healthy and unhealthy subjects (i.e. patients with e.g. an aneurysm). In a clinical scenario, it needs to be verified that denoising does not suppress important clinical information. This will be done in future studies based on this work.

Our study has several limitations. First, all networks were trained and applied on data acquired at the same scanner. In future work, we will train and test our approach on data from different scanners yielding different image characteristics. Second, the MRA dataset used in this study yields exclusively images from healthy subjects. In future work, we will apply the networks also to subjects with vessel abnormalities. MR images of human subjects also suffer from motion artifacts. The proposed uPL might also be a promising candidate for training networks for MR motion correction what will be investigated in future work. All in all, our results demonstrate that the untrained Perceptual Loss can be used in image denoising networks. To what extent it can be used for image enhancement in a clinical scenario, needs to be determined in future work. However, the untrained perceptual loss should definitely considered when training neural networks on MR images displaying fine, line-like structures.’

3) I suggest adding a clear research objective and research questions in the introduction section and specify what the main research problem or hypothesis is addressed.

We thank the reviewer for this important point. We added the following part to the introduction: ‚ As displayed in Table 1, most recent studies addressing image denoising of MR brain or MRA images focus on the most adequate network architecture while using conventional loss functions such as L1 and/or SSIM-loss. Therefore, the aim of this work is to investigate if the uPL can be used beneficially for 3D images containing line-like structures. The research objective is to assess the impact of uPL network parameters on the results. We are especially interested if also small, simple networks can be used in the uPL.’

4) Please add more recent references. Certainly, there has been more recent (within the last two years) research on this topic published in information science and/or computer science outlets. An academic search on the topic (using keywords from the manuscript's title) shows that there is recent work in this area. Therefore, authors must update their literature review.

Thank you for this comment. We added a Table including recent work (Table 1).

5) Starting from the previous works, I suggest introducing a table to summarize the most recent works and to highlight the novelty of the proposed work.

Thank you for pointing this out. We added a part to the introduction (see answer to question 3).

6) The manuscript only provides the structure of network, it is necessary to explain why these networks can work effectively and to show the concrete parameters.

---

## [Decision Letter · Decision Letter 2]

27 Jan 2025

Untrained Perceptual Loss for image denoising of line-like structures in MR images

PONE-D-24-26390R2

Dear Dr. Pfaehler,

We’re pleased to inform you that your manuscript has been judged scientifically suitable for publication and will be formally accepted for publication once it meets all outstanding technical requirements.

Kind regards,

Khan Bahadar Khan, Ph.D

Academic Editor

PLOS ONE

Additional Editor Comments (optional):

Reviewers' comments:

Reviewer's Responses to Questions

**Comments to the Author**

1. If the authors have adequately addressed your comments raised in a previous round of review and you feel that this manuscript is now acceptable for publication, you may indicate that here to bypass the “Comments to the Author” section, enter your conflict of interest statement in the “Confidential to Editor” section, and submit your "Accept" recommendation.

Reviewer #3: All comments have been addressed

Reviewer #4: All comments have been addressed

2. Is the manuscript technically sound, and do the data support the conclusions?

Reviewer #3: Yes

Reviewer #4: Yes

3. Has the statistical analysis been performed appropriately and rigorously? 

Reviewer #3: Yes

Reviewer #4: Yes

4. Have the authors made all data underlying the findings in their manuscript fully available?

Reviewer #3: Yes

Reviewer #4: Yes

5. Is the manuscript presented in an intelligible fashion and written in standard English?

Reviewer #3: Yes

Reviewer #4: Yes

6. Review Comments to the Author

Reviewer #3: I think the authors have addressed my comments and concerns adequately. The conclusions seem reasonable given the methods and results.

Reviewer #4: (No Response)

7. PLOS authors have the option to publish the peer review history of their article (what does this mean?). If published, this will include your full peer review and any attached files.

Reviewer #3: No

Reviewer #4: No

---

## [Editor Report · Acceptance letter]

PONE-D-24-26390R2

PLOS ONE

Dear Dr. Pfaehler,

I'm pleased to inform you that your manuscript has been deemed suitable for publication in PLOS ONE. Congratulations! Your manuscript is now being handed over to our production team.

Kind regards,

on behalf of

Dr. Khan Bahadar Khan

Academic Editor

PLOS ONE